# CycleTrans: a transformer-based clinical foundation model for safer prescription

**Yuhan Zheng[2,3], Xiaotao Lin[2,4], Kexuan Chen[1,2], Shengxin Zhu[1,2]**

[1]Research Centers for Mathematics, Advanced Institute of Natural Science,
Beijing Normal University, Zhuhai 519087, P.R.China
[2]Guangdong Provincial Key Laboratory of Interdisciplinary Research and Application for Data Science,
BNU-HKBU United International College, Zhuhai 519087, P.R. China
[3]School of Informatics, The University of Edinburgh
[4]Department of Mathematics, The University of Hong Kong
Shengxin.Zhu@bnu.edu.cn

## Abstract

Deep learning techniques are extensively utilized in prescribing drug combinations, drawing on extensive electronic medical records (EMRs). A prescription assistant may be able to provide immediate guidance on drug combinations for some urgent clinical situations. A well-controlled drug-drug interaction (DDI) rate and high recommendation precision are of great importance for a safe prescription. A lower DDI often implies the set of drug combinations should be as small as possible, which is challenging because EMR prescriptions for certain symptom(s) are often highly noised due to the diversity side symptoms of individuals. We propose a model comprised of cycle transformers (CycleTrans) to handle these challenges. CycleTrans employs cross-attention and transformers, integrates patients' longitudinal EMRs, enhances knowledge representations through the so-called cycle-embedding module, and thus predicts safer and more essential drug combinations for new-coming cases. The new model achieves the state-of-the-art in three dimensions: high precision (89%), low DDI rate (0.34%), and small drug set size (3.02) on the MIMIC-III benchmark dataset, surpassing previous bests of 73%, 5%, and 17 in each dimension, respectively. Such a significant advancement makes a much safer clinic prescription possible. The idea of the cycle transformer we proposed has considerable potential for other domains besides clinics, such as set recommendations, translation, and unsupervised representation learning in knowledge graphs.

## Introduction

Foundation models (FMs), a recent innovation in machine learning, are trained on vast datasets across multiple domains. They exhibit impressive performance in addressing various real-world AI challenges (Bommasani et al. 2022; Moor et al. 2023). Among the most notable models is OpenAI's ChatGPT, a predominantly text-based large language model (LLM) that has extended its capabilities to include image processing and data analysis (Eysenbach 2023). Remarkably, ChatGPT has demonstrated the capability to achieve passing scores in the United States Medical Licensing Examinations (OpenAI 2023). This accomplishment hints at its readiness in diverse fields such as clinical environments (Thirunavukarasu et al. 2023a; Kung et al.

2023; Thirunavukarasu et al. 2023b) and clinical foundation models (Zhang et al. 2023a; Steinberg et al. 2023).

LLMs like ChatGPT, though trained on general datasets, have proven effective in medical examinations (Robert 2023). However, success in passing written examinations does not necessarily imply clinical competence (Ayers et al. 2023). The clinical requirements for experience, timeliness, accuracy, and supervision pose significant challenges for the application of medical foundation models. There is currently no paradigm that adequately addresses the application of these models in the medical field, making their practical deployment contentious. Multiple articles have pointed out that these foundation models should be utilized as tools under supervision (Wornow et al. 2023). Our study is predicated on developing a clinical foundation model that matches patients with appropriate medications based on their specific symptom information. This approach aims to achieve precision, safety and expedited healing for patients.

Existing models, such as GatorTron and the Biomed-CLIP model, offer valuable insights for our research (Zhang et al. 2023b; Yang et al. 2022). Meanwhile, a plethora of studies utilizing the MIMIC-III dataset provide extensive analysis and benchmarks (Shang et al. 2019; Wang et al. 2019). Currently, an overemphasis on better predictive performance in many articles neglects a broader set of crucial factors. These include less labeled data, simplified model deployment, emergent clinical applications, multimodality, and novel human-AI interfaces (Wornow et al. 2023).

Not only do the results and model factors play a crucial role, but the dataset also significantly impacts the overall framework. To address the limitations of general LLMs in clinical practicality and the existing gaps in a comprehensive analysis of dataset and evaluation standards in MIMIC-III, a more nuanced approach is required. MIMIC-III is particularly noteworthy as it encompasses a vast collection of nearly 2 million patient notes from 2001 to 2012, documented in the ICU of the Beth Israel Deaconess Medical Center (Johnson et al. 2016). In the fast-paced environment of the ICU, quick decision-making is as essential as precision. In such settings, a minor oversight can lead to severe consequences, making manual evaluation crucial. This aspect of urgency and precision in real-world decision-making is often overlooked in many high-accuracy applications of the MIMIC-III model (Wornow et al. 2023).

To more effectively address the challenges previously outlined, we introduce the CycleTrans model. The principal contributions of this work are as follows:

1. This work has developed the CycleTrans model to predict specific medications for patients, according to their disease diagnoses. The model introduces a cycle-embedding module that enhances symptom and drug embeddings, which can be developed as a foundational model for many downstream tasks, such as "Clinical Trial Matching" and "Treatment Recommendation".

2. Cross-attention and transformers are employed to integrate patients' longitudinal data, alongside a drug attention mapping matrix for effectively mapping drug interactions.

3. The model achieves a state-of-the-art precision of **89.26%**, a low DDI rate of **0.34%**, and a minimum main drug set size of **3.02**.

## Related Works

Among deep learning approaches to drug recommendation, the most classical implementation is the instance-based approach, which focuses solely on the current health state information. Representative models include SMR (Gong et al. 2021) and LEAP (Zhang et al. 2017). For example, LEAP deals with label dependency and label instance mapping by combining a recursive decoder and content-based attention. However, such an approach does not incorporate well the information contained in the patient's historical health records (Wu et al. 2022). On the other hand, longitudinal drug recommendation methods can utilize the time dependency implicitly with longitudinal patient records (Le, Tran, and Venkatesh 2018; Yang et al. 2023). Based on this idea, many models have also begun to consider the effect of DDIs on the outcome of drug recommendations, thus making drug recommendation models more reliable (Wang et al. 2021b; Kim et al. 2023; Wang et al. 2021a; Yang et al. 2021). For example, GAMENet (Shang et al. 2019) is building graph models based on the co-occurrence of drugs in EMR through memory networks and graph neural networks. COGNet (Wu et al. 2022) is determining whether predictive medications require a new drug or merely follow the medical history based on the patient's historical health records and the DDI matrix. MEGACare (Wu et al. 2023) discerns complex relationships in the data by constructing an EMR hypergraph. Most of the existing longitudinal models conform to the encoder-decoder architecture, where first the encoder generates patient-level representations of known medical and patient data, and the decoder performs drug recommendation based on the embedding of the information(Wu et al. 2022). For more references, the reader can refer to (Ali et al. 2023).

## Methodology

In clinical trials, drug recommending requires accurate diagnosis, comprehensive knowledge of drug effects and interactions, and consideration of patient preferences and characteristics. Our model consists of two key components as shown in Figure 1, and a new fusion loss in (6) to capture all the information, enhancing the precision and reducing the DDI rate. The whole framework is shown in Figure 1.

In this model, Symptom and drug sets are treated akin to words and thus ideas for translation such as cycleGAN (Zhu et al. 2017) can be borrowed. However, for a clinic emergency, drug combination recommendation should be as safe as possible and higher precision is desired. Therefore, we design a new loss such that high precision and low DDI can be achieved.

## Symptoms and Historical Diagnosis Fusion Module

This module is referred to as the "Drug Transformer" in *Figure 1*. Considering that the procedure recommendation must account for patient preferences and characteristics, including historical data, a fusion module employing a multi-head attention mechanism has been designed (Vaswani et al. 2017):

$$z_d^i = Attention(h_t W_q^i, h_{1:t-1} W_k^i, h_{1:t-1} W_v^i) \quad (1)$$

where $z_d^i \in \mathbb{R}^{N \times \frac{D}{h}}$ represents the response from the $i^{th}$ attention head, managing $N$ symptom types across $h$ total heads. The term $h_t$ denotes the current diagnosis, while $h_{1:t-1}$ embodies the historical diagnosis data. Additionally, $W_q^i, W_k^i, W_v^i \in \mathbb{R}^{D \times \frac{D}{h}}$ are designated as the weight matrices for the $i^{th}$ query, key, and value within the $D$ dimensional embedding space of the multi-head attention mechanism.

Finally, we concatenate and transform $h$ heads to form the representation of diagnosis $z_d$ by

$$z_d = Concat(z_d^1, \ldots, z_d^h)W_o, \quad (2)$$

where $W_o \in \mathbb{R}^{D \times D}$ denotes the weight matrix that integrates the final output, while $Concat(z_d^1, \ldots, z_d^h) \in \mathbb{R}^{N \times D}$ represents the concatenated outputs of the $h$ attention heads. Subsequently, a cross-attention mechanism is applied, utilizing historical diagnoses as the "key" and current symptoms as the "query", to merge this information for a certain patient. This approach enables the model to assimilate comprehensive data, encompassing both the doctors' current diagnosis and historical patient information.

## Cycle-embedding Module

As this system assists doctors in decision-making, it is important to improve the predicting precision of the key drugs, and other minor drugs can be left to the doctors' discretion based on the possible drug interactions and the patients' conditions. Consequently, this approach allows the system to offer more precise and tailored recommendations for optimal treatment strategies.

The sets of symptoms and drugs are conceptualized as two distinct distributions, with a Cycle-embedding module, inspired by CycleGAN (Zhu et al. 2017), facilitating the primary model's training. Minor drugs are considered random perturbations, while key drugs represent the primary distribution. The objective is to transform one distribution into another, akin to a translation task where an accurate translation between languages should be reversible.

The "Symptom Transformer", as depicted in Figure 1, represents the cycle assisting module. This module processes the currently inferred drug as input, converting it back

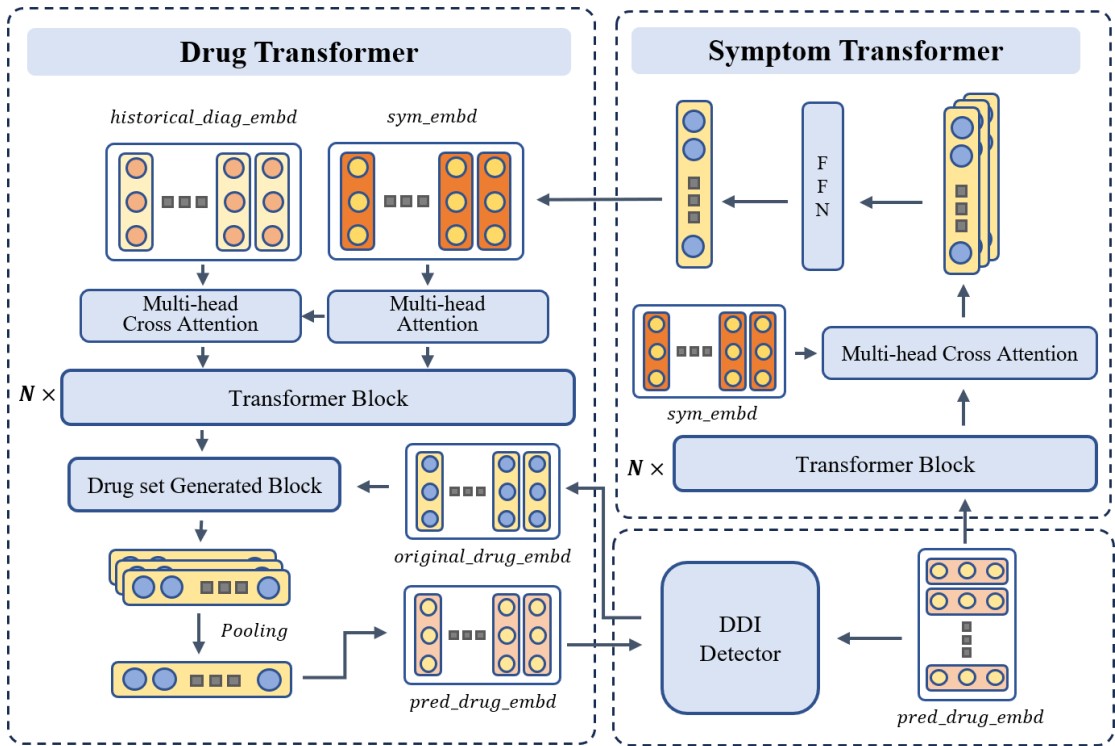

Figure 1: The CycleTrans architecture comprises two key modules: the Drug Transformer Module, which suggests drugs, and the Symptom Transformer Module, which converts predicted drugs back into symptoms for cycle checking, thereby improving the embeddings' quality for both symptoms and drugs.

into symptoms to enhance embedding quality. The associated loss is quantified by the distance between actual and predicted symptoms:

$$\mathcal{L}_{cycle} = \frac{1}{N} \sum_{n=1}^{N} \|h_s^n - \hat{h}_s^n\|, \qquad (3)$$

where $h_s^n$ is the embedding of the $n^{th}$ patient's symptoms. $N$ denotes the number of samples.

**Safe Drug Loss Design**

In light of the need for high precision and a low DDI rate, the DDI loss has been restructured, drawing inspiration from 4SDrug (Tan et al. 2022), which also expedites the training process.

$$\mathcal{L}_{ddi} = \frac{1}{N} \sum_{n=1}^{N} p_n \cdot M_{DDI}, \qquad (4)$$

where $M_{DDI}$ is the adjacent matrix of DDI, and the $p$ is the prediction of drugs set for the $n^{th}$ patient. In addition to the DDI loss, it also requires a set classification loss to enhance the precision:

$$\alpha \mathcal{L}_{cls} = \alpha_1 \mathcal{L}_{CrossEntropy} + \alpha_2 \mathcal{L}_{EMD} \qquad (5)$$

where the $\mathcal{L}_{EMD}$ is the earth mover's distance loss that measures the difference that a set $A$ transfers to a set $B$.

Combining the above three parts, the final loss is formulated as follows:

$$\mathcal{L} = \alpha \mathcal{L}_{cls} + \beta \mathcal{L}_{cycle} + \gamma \mathcal{L}_{ddi}, \qquad (6)$$

where $\alpha, \beta$, and $\gamma$ are hyperparameters that control the relative importance of different objectives, such as reducing the DDI rate or increasing the precision.

| Datasets | MIMIC III | | |
|---|---|---|---|
| **Model (year)** | **Precision(%)** | **DDI rate** | **Avg #** |
| LEAP (17) | 65.49±0.33 | 0.073±8e-4 | 18.71±0.07 |
| GAMENet (19) | 76.31±0.30 | 0.086±6e-4 | 27.21±0.11 |
| SafeDrug (21) | 76.47±0.25 | 0.059±5e-4 | 19.92±0.16 |
| 4SDrug (22) | 76.04±0.16 | 0.054±4e-4 | 14.64 ± 0.07 |
| SHAPE (23) | 79.06 ± 0.09 | 0.068±3e-4 | 20.99±0.12 |
| ACDNet (24) | 79.04±0.21 | 0.086±1e-3 | 20.49±0.12 |
| **CycleTrans** | **89.24±2e-3** | **0.008±8e-3** | **2.81±0.80** |

Table 1: Experimental results. Bold and underlined texts indicate the best and the second-best scores. Avg # refers to the average number of drugs used in a case. Results of the above models except CycleTrans are from the SHAPE and ACDNet papers. (Liu et al. 2023a; Mi et al. 2024)

## Results

The experiment utilized a composite loss function with weights assigned as follows: cycle loss at 0.2, classification losses at 0.3 and 0.02 ($\alpha_1 = 0.3$, $\alpha_2 = 0.02$), and DDI loss at 0.1. The RAdam optimizer was employed with an initial

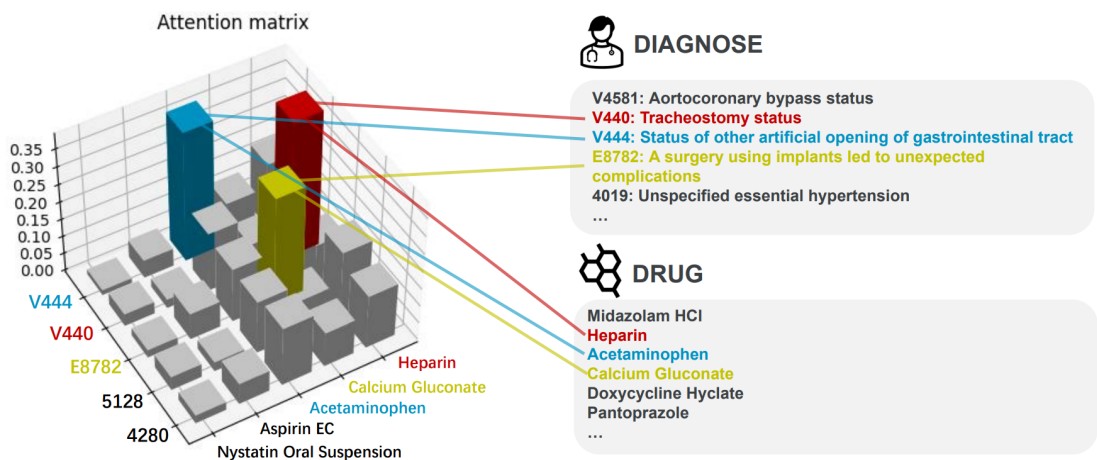

Figure 2: Attention matrix of drug-diagnosis correlations applied to visualization of predictions. The left side shows a three-dimensional histogram of the attention matrix, with the height indicating the size of the weights, while the right side shows the predicted sample (HADM_ID: 167243).

learning rate of 5e-4, and training spanned 20 epochs with a batch size of 50. The results are presented in Table 1.

In the results presented in Figure 2, we illustrate an example of an attention (sub)matrix applied to model prediction, where the 3d bar graph on the left represents the attention weights, which indicate the relevance of certain diagnoses and drugs. The right part represents a prediction example where diagnoses are known inputs and drugs are model-predicted outputs. For instance, the status of other artificial openings of the gastrointestinal tract (V444), which involves surgically-created openings in the digestive system, shows a significant correlation with Acetaminophen in the model's predictions. This suggests that the model emphasizes the management of post-surgical pain, a condition that can be alleviated by Acetaminophen (U.S. Food and Drug Administration 2023). The CycleTrans not only shows a high capacity of capturing the relationships between diagnoses and medical treatments, but also offers a view of the factors influencing its predictions.

## Discussion

In this paper, we have developed CycleTrans for utilizing the extensive MIMIC-III corpus. Our model excels in multiple dimensions, achieving a high clinical precision rating of 89.26%, a low DDI rate of 0.34%, and a minimum main drug set size of 3.02.

There are other standards that CycleTrans can improve, such as evaluation by medical professionals. The traditional NLP metrics have been shown to correlate poorly with human judgments (Reiter 2018; Hu et al. 2022; Liu et al. 2023b). Despite recognizing the need for diverse evaluative standards, especially in terms of precision and rapid response required in medical and clinical settings, we find that larger, domain-specific pre-trained models (e.g. GatorTron) excel in modeling longer phrases and identifying semantic categories (Zhang et al. 2023b). However, for complex NLP tasks like clinical reasoning judgments and specialized med-

ical questions, even LLMs like GatorTron struggle to discern key information from longer paragraphs. Similarly, our model also necessitates additional data, particularly recent clinical domain data, to substantiate and validate.

There are follow-up questions worth considering. Not just for Clinical FMs, but across neural network models, still lack clear AI explainability. Transferring these models to the medical field for analysis and evaluation doesn't suffice to understand their true practical value. Additionally, ethical and moral concerns about AI-generated conclusions remain a critical topic. LLMs' outputs are increasingly preferred for their quality and empathy, even when compared to responses from real doctors on social media (Ayers et al. 2023). Furthermore, clinical foundation models like ClinicalBERT, Med-PaLM 2, and GatorTron have even exceeded the capabilities of these general LLMs (Singhal et al. 2023). However, in all medical disciplines, interpersonal communication is a vital component of patient care. LLMs have been proven to replicate existing biases and are prone to disseminating incorrect information and perpetuating errors in AI decision-making (Clusmann et al. 2023). How to interpret Clinical FMs in a way that ensures they do not produce ethical biases remains an unresolved issue. Safety, efficacy, and ethical concerns remain unresolved. Enhancing the transparency and explainability of models is imperative in medicine to foster understanding, trust, and effective management among users of these systems (Clusmann et al. 2023). Undoubtedly, both LLMs and Clinical FMs are transforming the fields of medicine and clinical practice.

In the future, we will consider larger datasets, such as the MIMIC-IV dataset, as well as more other diverse multimodal datasets, to further pretrain our model enhancing its robustness.

# Acknowledgement

The authors would like to thank support from the Interdisciplinary Intelligence Super Computer Center of Beijing Normal University at Zhuhai. This work was partially supported by the Natural Science Foundation of China (12271047); UIC research grant (R0400001-22; UICR0400008-21; UICR04202405-21); Guangdong College Enhancement and Innovation Program (2021ZDZX1046).

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

# Appendix

## Definitions

- **Precision**

  Precision denotes the proportion of the intersection of model-recommended drugs and real-labelled drugs to the model-predicted drugs. the higher the value of precision, the more accurate the model-recommended drugs are. In this definition, we use the average Precision of all samples as a judgement criterion, so there exists a process of finding the mean. The calculation method of **Precision** is shown as follows:

$$\frac{1}{N} \sum_i^N \frac{|\boldsymbol{p^{(i)}} \cap \boldsymbol{D^{(i)}}|}{|\boldsymbol{D^{(i)}}|}$$

  where $p^{(i)}$ denotes the predicted drug set of the model, $D^{(i)}$ the number of drugs in the basic real drug set, and $i$ denotes the index of the test drug set.

- **Drug-Drug-Interaction rate (DDI rate)**

  Drug-drug interactions (DDIs) refer to the phenomenon where two or more drugs interact with each other and negatively affect the way they work in the body. This can lead to a variety of outcomes such as reduced efficacy, increased side effects, or even toxicity of a particular drug. In our work, we define the DDI rate as a measure that implies the proportion of drug combinations provided by the model that result in a DDI situation. The **DDI rate** can be calculated by the following method:

$$\frac{1}{N \sum_{x,y} \mathbf{1}} \sum_i^N \left| \left\{ (d_x, d_y) \in \left( \boldsymbol{D^{(i)}} \,\&\, \boldsymbol{\mathcal{E}}_{ddi} \right) \right\} \right|$$

  In the formula above, each drug pair $(d_x, d_y)$ will be counted in the set if this pair exist in the drug knowledge base, which represented as $\mathcal{E}_{ddi}$.

## Source code

The source code can be found on the repository[1].

---

[1] https://github.com/Undefeated-man/Cycletrans/blob/main/README.md