# OpenReview forum: "CycleTrans: a transformer-based clinical foundation model for safer prescription"
_AAAI.org/2024/Spring_Symposium_Series/Clinical_FMs — AAAI 2024 SSS on Clinical FMs_

### Official Review · Reviewer_CRjH · 2024-02-13

**Rating:** 6
**Confidence:** 4

**Review:**

This paper presents an interesting and impactful approach to improving drug recommendation systems and reducing drug drug interactions.

I would be happy to raise my score if the weaknesses and questions are addressed.

Strengths:
1.  The paper is well-organized and was very enjoyable to read.
2. The idea appears to be novel and methods are thoroughly explained
3. Figure 2 is a clinically relevant and easy to interpret example that demonstrates the utility and transparency of this approach to assist healthcare workers. It would be nice to see a similar example of how DDI are assessed for the predicted medications.

Weaknesses:
1. Potential harm of drug-drug interactions and expected benefits of drug recommendation systems are not discussed. A couple sentences would suffice. For example, statistics regarding prescription errors and effects would help emphasis the importance of this research area.
2. Paper does not explain novel contributions compared to prior work, particularly longitudinal drug recommendation methods that consider the effect of DDIs. I am assuming it is the use of cross-attention, transformer architectures, and a cycle-embedding module for healthcare but this should be explained as gaps in prior work. New loss function could also be referenced as a contribution.
3. Results demonstrate impressive gains over baselines but do not include confidence intervals. Including averages across multiple runs or random seeds is important to convey significance of results. If there is a reason this is not feasible, then that should be mentioned.
4. Diagram is fairly clear but duplicate inputs and labels, such as for 'drug_embd' and 'sym_embd', in the drug transformer and symptom transformer is confusing and makes diagram more difficult to parse than it need be. I would recommend reworking the figure to reference common inputs once outside of the modules. Also there are two colors for drug_embd used (blue and pink). One color should be used for the same input. The acronyms and colors should be explained in the image description.
5. DDI rate is not explained concretely. This is a very important term in pharmacology so more information regarding how the DDI detector is crucial.
6. More information regarding the classification task is needed. For example, use of the precision metric implies some score threshold is set to convert probabilistic outputs.

Questions:
1. How is DDI rate measured? The author mentions that "A lower DDI often implies the set of drug combinations should be as small as possible". Is set refer to a specific combination or number? Drug drug interaction rate should measure the extent to which drugs interact and not the number of drugs. Assuming the number of drugs given is a proxy for drug interaction rate is not a valid assumption. Can the author elaborate on their assessment of DDI and provide references if possible?

Other:
1. It seems that predicted drug predictions learn from historical data/prescriptions. It would be interesting to see extensions of this work that also consider patient outcomes in the prediction. The model may suggest alternative medications not administered, although these predictions would need to be verified by medical professionals.

---

### Official Review · Reviewer_Sbmp · 2024-02-14

**Rating:** 6
**Confidence:** 4

**Review:**

The paper proposes a deep learning framework that integrates a cycle transformer inspired by CycleGAN and a DDI loss inspired by 4SDrug to predict the drug combination prescription. The experiment carries out with MIMIC-III dataset, indicating that the proposed model outperforms previous models by large in terms of Precision, DDI rate, and Average number of drugs in the set.

### Strength:
1. The experimental results are promising.
2. The model design seems reasonable.
3. The problem of predicting drug combination is important.

### Opportunity to improve:
1. The reference number of Baseline in Table 1 does not align with other citation styles.
2. The definition of Precision needs clarification. If multiple drugs suit the patient, does hit with any of them count as 1? If it is not the case, it would be very unusual to see the Precision outperforming baselines by this far while maintaining such a small Avg #. It is extremely crucial to clarify this.
3. How is the DDI adjacent matrix defined? Without clarification, it can be very confusing. There may raise concerns if the definition of DDI is actually making sense in terms of medicine and for clinicians.
4. I do not see why L_{EMD} can be jointly added to the loss function, even though WGAN uses it. What are set A and set B, and why is there a need to transfer A to B?
5. Hyperparameters tuning will be very important since there are multiple loss functions. I assume adding appendices to explain how to reproduce the results and/or code would be helpful.
6. I do not think it is a good idea to call it a “foundation model”, since it has so few downstream tasks. It is not necessary to name it a foundation model just to fit in the symposium topic.

---

### Official Review · Reviewer_23UG · 2024-02-17
**The document introduces CycleTrans, a transformer-based model for safer prescription recommendations, addressing challenges in clinical reasoning and medication prediction, achieving high precision and low drug-drug interaction rates.**

**Rating:** 9
**Confidence:** 4

**Review:**

The paper presents a study on the development of the CycleTrans model for predicting specific medications for patients based on their disease diagnoses. The model incorporates a cycle-embedding module to enhance symptom and drug embeddings, utilizes cross-attention and transformers to integrate patients' longitudinal data, and achieves high clinical precision and low drug-drug interaction (DDI) rate. The study also discusses the need for additional data, ethical concerns, and the unresolved issue of AI explainability in the medical field.

Pros:

The study introduces a novel model, CycleTrans, which addresses the need for precise medication recommendations based on patient diagnoses.

The incorporation of a cycle-embedding module and the use of cross-attention and transformers demonstrate a comprehensive approach to addressing the complexities of medication recommendation in clinical settings.

The model achieves high clinical precision and low DDI rate, indicating its potential for improving patient safety and treatment efficacy.

Cons:

The study acknowledges the need for additional data, particularly recent clinical domain data, to substantiate and validate the findings, indicating potential limitations in the current model's training and evaluation.

Ethical and moral concerns about AI-generated conclusions and the lack of clear AI explainability in the medical field remain unresolved, raising questions about the practical application of the model in real-world clinical settings.

The study does not provide a detailed discussion of potential biases or limitations in the model's predictions, which could impact its practical significance and real-world applicability.

Overall, the study presents a novel approach to medication recommendation in clinical settings, but it also highlights the need for further validation, consideration of ethical concerns, and addressing potential biases in the model's predictions. The work's significance lies in its potential to improve patient care and treatment outcomes, but its practical application may be limited by unresolved ethical and explainability concerns.

---

### Official Review · Reviewer_P3W5 · 2024-02-21
**CycleTrans, a transformer based model for safer prescription assistant**

**Rating:** 4
**Confidence:** 3

**Review:**

Summary of Contributions:

The work provides a transformer based method to create a “prescription assistant” by training on longitudinal EMRs. The authors propose CycleTrans to learn good representations which surpasses performance achieved by prior works.


Strengths:
1. The results are excellent when compared to prior works.
2. The block diagram is good representative of the method described in the paper. The visualization in the final page is also quite impressive.
3. Hyper-parameters and training details are provided in the paper.
4. The EMD loss strategy to increase precision is a good idea.


Weaknesses:
1. The biggest concern is that, the work authors propose might not a foundation model in the traditional sense. In this work, the generalised pre-training phase and domain specific fine-tuning phase wasn’t clear from the manuscript.
2. The dataset MIMIC-III is quite small for training a foundation model. Maybe MIMIC-IV even some synthetic datasets like Synthea might have helped the model learn good quality representations in the pre-training phase.
3. Experimental results have been demonstrated on just a single dataset (MIMIC-III). The generalizabilty of the method cannot be verified unless other benchmark datasets are also used.